# Supporting Medical Staff from Psycho-Oncology with Smart Mobile Devices: Insights into the Development Process and First Results

**DOI:** 10.3390/ijerph18105092

**Published:** 2021-05-11

**Authors:** Johannes Schobel, Madeleine Volz, Katharina Hörner, Peter Kuhn, Franz Jobst, Julian D. Schwab, Nensi Ikonomi, Silke D. Werle, Axel Fürstberger, Klaus Hoenig, Hans A. Kestler

**Affiliations:** 1Institute of Medical Systems Biology, Ulm University, 89081 Ulm, Germany; johannes.schobel@hnu.de (J.S.); madeleine.volz@uniklinik-ulm.de (M.V.); julian.schwab@uni-ulm.de (J.D.S.); nensi.ikonomi@uni-ulm.de (N.I.); silke.kuehlwein@uni-ulm.de (S.D.W.); axel.fuerstberger@uni-ulm.de (A.F.); 2DigiHealth Institute, University of Applied Sciences Neu-Ulm, 89231 Neu-Ulm, Germany; 3Department of Psychosomatic Medicine and Psychotherapy, Ulm University, 89081 Ulm, Germany; katharina.hoerner@uniklinik-ulm.de (K.H.); klaus.hoenig@uniklinik-ulm.de (K.H.); 4Comprehensive Cancer Center, Ulm University Hospital, 89081 Ulm, Germany; peter.kuhn@uniklinik-ulm.de; 5Strategic IT, Ulm University Hospital, 89081 Ulm, Germany; franz.jobst@uniklinik-ulm.de

**Keywords:** data collection, smart mobile devices, development process, psycho-oncology

## Abstract

Cancer is a very distressing disease, not only for the patients themselves, but also for their family members and relatives. Therefore, patients are regularly monitored to decide whether psychological treatment is necessary and applicable. However, such monitoring processes are costly in terms of required staff and time. Mobile data collection is an emerging trend in various domains. The medical and psychological field benefits from such an approach, which enables experts to quickly collect a large amount of individual health data. Mobile data collection applications enable a more holistic view of patients and assist psychologists in taking proper actions. We developed a mobile application, FeelBack, which is designed to support data collection that is based on well-known and approved psychological instruments. A controlled pilot evaluation with 60 participants provides insights into the feasibility of the developed platform and it shows the initial results. 31 of these participants received paper-based questionnaire and 29 followed the digital approach. The results reveal an increase of the overall acceptance by 58.5% in the mean when using a digital screening as compared to the paper-based. We believe that such a platform may significantly improve cancer patients’ and relatives’ psychological treatment, as available data can be used to optimize treatment.

## 1. Introduction

With the improvement of general living conditions and increase of life expectancy, oncologic diseases have become increasingly prevalent in the population. Besides, an increasing number of patients survive cancer or live with the disease for long periods of time [1,2]. This not only affects patients physical health, but also their and their families psychologic well-being [1,3]. In fact, psychological stress factors have been reported to influence cancer insurgence and development [4,5]. In addition, psychological stressors also affect patients’ outcome and quality of life during chemotherapy and, therefore, they need to be taken under strict control during treatment [1,6]. In this line, 30–60% of cancer patients report psychological issues when dealing with their cancer disease or during their treatment [7]. To be able to maintain an individual level of quality of life, patients are psychologically looked after and treated (if necessary and applicable). This inter-disciplinary field is called psycho-oncology, and it mainly deals with the psychological, social, or ethical facets of cancer patients [8]. It accompanies patients as well as their family members during all stages of cancer, and it helps with aspects that may influence the process of the disease in a positive or negative way. Nowadays, psycho-oncology is a well-recognized field and established in many hospitals, as it is seen as an integral service for patients and relatives. Moreover, cancer counseling centers exist that may be attached to hospitals offering an anonymous consultation or assistance in certain situation. During such consultations, psychologists may identify stress factors and provide advice how to deal with the latter through private counselings or group therapies. Stress is a well-known aging fact [9,10], and such stress factors may change over time (i.e., as the cancer disease progresses). Furthermore, patients may develop a need for psychological counselling as their current situation changes. In order to properly detect such ever-changing needs and requirements, patients, as well as their relatives, should continuously process structured screening instruments (i.e., every week) [11]. This fact is also suggested by international treatment guidelines, where such screenings are seen as one fundamental pillar to identify whether a patient requires psychological assistance or treatment. Such screening instruments, in general, are mostly self-report questionnaires assessing different factors of every day life. For example, one dimension may deal with the general health condition (e.g., pain; [12,13,14]), whereas others may deal with daily life situations (e.g., financial situation, traffic, or work day; [15]). The Hospital Anxiety and Depression Scale (HADS), European Organisation for Research and Treatment of Cancer (EORTC QLQ C30), Hornheider Screening-Instrument (HSI), or the Distress Thermometer (DT) are common instruments in psycho-oncology, as they are short, available in various languages and they have proven their feasibility in multiple studies [16,17].

Unfortunately, such psychological instruments are still mostly carried out using paper-based questionnaires. Naturally, this comes with a lot of drawbacks, like printing and transporting the documents. Further, the questionnaire has to be digitalized in order to make the results available for subsequent evaluation. Moreover, the logic of the instrument has to be put into text, relying on patients to properly process respective questionnaire (i.e., *“If you answered this question with ‘yes’, please continue with question X, otherwise continue here.”*) [18] found out that approximately 50% of costs that are related to collecting and processing data in clinical trials could be saved when relying on digital instruments instead of paper-based ones, which is also beneficial for personalized medicine [19]. This would also increase data quality [20], reduce the time that is required to collect such data [21], while still producing valid and comparable results [22]. Using such digital solutions in various application scenarios is increasingly demanded by domain experts.

Our current research project aims at supporting psychologists and medical personnel in collecting data from cancer patients. More specifically, a platform, called FeelBack, was developed that allows psychologists to collect data from cancer patients using smart mobile devices (tablets and smartphones). In a first step, the Distress Thermometer was digitized and implemented within the platform. The latter is a well-established and standardized instrument applied in cancer research worldwide [23,24,25].

A pilot feasibility study with multiple participants (i.e., cancer patients and relatives) was conducted in cooperation with a local cancer counseling center to validate and investigate the applicability our implementation. The data collected are used to evaluate whether such a platform is feasible for the respective scenario and participants would use such digital solutions.

## 2. Related Work in mHealth and Psycho-Oncology

Related work in the context of digitization of medical processes or medical applications, in general, are manifold. We identified two categories of related work that are particularly relevant in the context of this manuscript. First, we discuss mobile data collection applications in medical or psychological scenarios in general. Subsequently, we relate our work to other research projects from (psycho-)oncology.

### 2.1. Related Work in the Context of Mobile Data Collection and mHealth Applications

Over the last 15 years, several projects successfully implemented mobile applications in the medical domain. In [26], the benefits of such a mobile data collection approach are discussed in great detail. A large-scale survey with physicians revealed the role of novel technologies (i.e., eHealth and mHealth) and telemedicine in general from the physicians’ point-of-view [27].

Smartphones are increasingly used in medical scenarios to support either medical staff or patients. For example, [28] presents a mobile application for documenting wounds and, hence, improving the general workflows in clinics through digitization. The authors of [29] used tablet computers to enable patients indicate pain locations via a drawing module. The respective application presents a human figure from front and back. Different pain-types (i.e., cramping, stabbing) and intensity can be selected and visualized on a respective figure to better communicate with physicians in charge. The results showed that such digital pain drawings may significantly improve the doctor-patient communication, as the physicians’ understanding is improved dramatically. Other applications, like [30], apply concepts that are known from video games to help cancer patients to quit smoking. Patients report overall positive feedback and state, that the application helped them to deal with their cravings.

Research projects, like *Track Your Tinnitus* [31,32], *Manage My Pain* [33], or *PsychLog* [34] apply crowd sensing techniques to collect respective data from users in daily situations. Thereby, the mobile application notifies the user to fill in a screening assessing, for example, their current distress regarding tinnitus. Data are collected based on structured instruments (i.e., questionnaires) and submitted to a server. Depending on the application scenario, additional sensor data are also collected (i.e., the loudness of the surrounding environment, GPS position) and submitted. This may help researchers to address specific research questions. Finally, [35] provides lessons learned from three different mobile intervention studies.

In [36], a model-driven approach is presented that allows domain experts (i.e., doctors, psychologists) to design and implement instruments themselves. The models are executed within an engine that is capable of running on smart mobile devices [37,38]. A large-scale and long-running study showed the applicability of this approach [39] and measured mental effort when implementing such mobile data collection applications [40].

In the literature, there exist a variety of surveys that identified sophisticated mobile data collection frameworks [41,42]. Such frameworks and approaches are mature enough to have been successfully transferred from research projects to industry [43]. Although mobile data collection can be a powerful technique for supporting sophisticated scenarios in the medical domain, [44] discusses the limitations with respect to mobile data collection. Although mHealth applications supporting medical care are commonly acknowledged, there are still some open issues concerning their implementation. Healthcare providers recognize the relevance of this source of clinical data [45,46]. However, concerns are frequently raised regarding the clarity, transparency, and privacy issues of these applications [45]. The lack of secure and effective communication between the user and healthcare providers is frequently hampering these applications in healthcare routine. In this context, it is of utmost importance to select an appropriate approach for each given scenario. A study revealed that a significant amount of reviewed mHealth applications are of poor quality and lack data security concepts or transparency [47]. In this context, Zhou et al. [48] found that after download and installation, patients do not use mHealth applications because of security concerns, loss of interest, or hidden costs. Additionally, the participants reported that entering a large amount of data on smartphones may be cumbersome and scare them off. All in all, these criteria are summarized under the aspects of security, usability, privacy, appropriateness, suitability, transparency, and content, safety, technical support, and technology [49,50,51,52].

### 2.2. Related Work in the Context of Psycho-Oncology

In a more narrow and focused research area, other projects may also be of particular interest. In general, [53] describes the need for more psycho-oncological eHealth applications. The findings from a conducted study indicate that web-based applications are the most preferred medium for patients to obtain information with respect to their treatment or therapies. Further, the most relevant topics to be discussed are abilities to cope with anxiety, depression, and tips to improve the quality of life.

As another example, [54] illustrates a mobile stress management application for cancer survivors. The authors state that stress management interventions may improve the quality of life of patients. In order to provide such techniques to a broad audience, a smartphone application was developed that assesses the self-perceived distress of patients. Furthermore, the application provides different learning modules (i.e., what is stress, mindful breathing) to teach patients how to cope with their current situation.

Moreover, [55] discusses a psychological treatment that is based on the needs of cancer patients. Using standardized screenings for monitoring patients may result in additional overhead in hospitals. Furthermore, established work-flows may also need to be adapted to properly implement such data collection routines in practice. This overhead may be properly compensated when implementing digital screening instruments. The authors provide recommendations for developing sophisticated solutions. These recommendations include variables to be assessed by such screening instruments, required hardware devices, or actual implementation aspects.

The project *Screen2Care* is described in [56]. This platform also monitors psychological factors from cancer patients. However, the authors rely on their own screening instrument rather than using an existing, established instrument. A study evaluating the feasibility and applicability of the developed platform was carried out with patients and caregivers. The results indicate that patients react very positively to digital data collection. The results from caregivers and medical personnel was positive; however, some limitations could also be observed. For example, caregivers state that the digital data collection procedure takes too much time (i.e., up to 4 min.) as compared to the traditional paper-based method Additonally, work-flows have to be adapted to include such screenings in the daily routine. This is not always feasible for personnel due to a lack of time. It is noteworthy that the *Screen2Care* project tries to closely follow the guidelines described in [55].

Trautmann et al. [57] describe the development process of an electronic patient-reported outcome system. Thereby, patients have to fill in standardized instruments (i.e., Distress Thermometer, Quality of Life, …) on tablet computers to provide feedback regarding their own physical and mental condition. The platform was developed in close cooperation between IT experts, physicians, caregivers, and psychologists. Digitally collected data were attached to the electronic health record of respective patients. Again, patients and medical personnel rated the application as useful, as underlined by the study results.

## 3. Requirements for Developing a Psycho-Oncological Platform

In this section, the application scenario is presented. Therefore, the current as-is state of psycho-oncology in the local hospital is described and compared to the to-be state when implementing mobile data collection processes. In this context, the requirements from domain experts are highlighted. Finally, the architecture of the developed platform is described.

### 3.1. Application Scenario

The application scenario that is described in this manuscript follows the rules of the German psycho-oncological primary care (i.e., “psychoonkologische Grundversorgung”). In this context, the psychological needs and requirements of patients have to be recognized by medical personnel. Psychologists, in turn, must then provide adequate treatment for respective patients (i.e., psychological interviews, …). In this context, family members or close relatives of cancer patients are also covered by these rules.

Unfortunately, not all patients that should get psychological counseling are actually detected by respective personnel [55]. This may be because of various reasons: First, oncologists may need to spontaneously decide whether a patient needs counseling or not. Furthermore, this spontaneous judgment only considers the current situation and may not include judgments from previous meetings (i.e., last month). Moreover, patients may have issues talking to the oncologist regarding psychological problems. Often, patients visit cancer counseling centers, which may not be connected to a hospital. Consequently, psychological screening data assessed in such local counseling centers may not be shared with treating medical personnel in the hospital.

The process that is described in the as-is state has various limitations and flaws (cf. Figure 1). First, data are mostly collected using traditional paper-based questionnaires. To make these data available across different organization units, it they digitalized using a scanner and attached to a medical record as PDF document. Note that respective data most likely cannot be processed and evaluated automatically. Second, to obtain longitudinal information of patients, such screenings have to be performed continuously over an extended time span (i.e., weekly over 12 months). It may be the case that a patient currently does not require psychological treatment, but it will develop such needs a few months later, as the disease progresses. Finally, a patient can visit cancer counseling centers, hospitals, psychologists, or general practitioners for screening. However, there is no direct communication between the treating hospital and psychological facilities. Most likely, screening data that are captured in such counseling centers will not be shared, which results in an incomplete psychological profile of the respective patient.

### 3.2. Requirements for the FeelBack Platform

The following requirements (Req) could be extracted when conducting structured interviews with domain experts from the medical center at Ulm University. Note that this list only comprises the most important requirements.
Req-01 (Various Screening Instruments): an extendable framework should allow to add other well-established screening instruments (i.e., Hornheider Screening Instrument) at a later stage to deal with new requirements of caregivers.Req-02 (Mobile Data Collection): data should be collected in a digital manner by relying on smart mobile devices (i.e., smartphones or tablets). For this purpose, a mobile application has to be developed that is capable of visualizing respective instruments and storing collected data.Req-03 (Adaptive User-Interface): because the majority of the patients are elderly, a very simplistic user-interface has to be designed for the mobile application to not overburden the patients and streamline data input process.Req-04 (Passwordless Login): another login mechanism has to be provided, as most of the users do not have an e-mail address or may not want to share this information in the platform. Best-case, patients do not need to remember their own credentials (i.e., username and password).Req-05 (Automatic Evaluation): submitted screening instruments should be automatically evaluated based on given rules (i.e., cut-off values). The results should be displayed to assigned caregivers, which may use this information in psychological counselings.Req-06 (Visualize Results): the collected data should be visualized according pre-defined rules. This should help the caregivers to obtain an immediate overview of a patient. More specifically, the original data of a particular filled in screening instrument should be accessible by caregivers.Req-07 (Patient History): when using the application for a longer period (i.e., several months and ongoing), caregivers should be able to see the history of the patient. This history should visualize the self-assessed distress over selected period as line-charts.Req-08 (Data Export): it should be possible to export data collected in a an electronic data exchange format (i.e., HL7 FHIR compliant resources). This should allow for attaching the history to an electronic health record stored within a hospital information system (HIS).Req-09 (Split Patient and Medical Data): for security reasons, patient data (i.e., name or address) should not be stored in the same database as medical data (i.e., screening data). This should prevent leaking sensitive information. Data are only aggregated via the backend API, which should be secured via state-of-the-art techniques.Req-10 (Patient Empowerment): patients should be empowered to control the flow of their own data. For example, patients shall give access or revoke access to their data.

Some of these requirements (i.e., splitting patient and medical data) act as fundamental pillars in the to-be process and they are of utmost importance. The following paragraphs describe the new process to provide insights into the design process (see Figure 2).

The FeelBack platform will act as a centralized hub for collecting data that are related to psychological distress from cancer patients and their relatives. Thereby, various healthcare providers, like hospitals or cancer counseling centers, can be added. Patients, in turn, are registered when they first request psychological treatment or assistance. In this context, it does not matter whether they are visiting a hospital or a practicing psychologist. Patients may then fill in various instruments in order to provide data regarding their own condition or distress using smart mobile devices (i.e., tablets or smartphones). Data are then stored in the central hub and attached to their health record. Patients are never registered via their name or other identifying attributes (like e-mail address), ensuring an isolated and anonymous platform. Interviews with domain experts and patients revealed that this may be a critical requirement for patients. Participants, in turn, are solely identified via an ID card that will be handed out at their first visit. Hence, all of the data collected are attached to a pseudonym. Because of this fact, the FeelBack platform itself cannot be directly connected to any hospital information system, hence a patient cannot be mapped automatically based on personal details.

Screening data that have been collected when visiting different healthcare providers are then shared among the latter. Patients, in turn, shall be empowered to control which provider can see respective information and revoke access if required. Collecting information over a certain time span will result in a detailed history of psychological issues that are related to one particular participant. Psychologists, in turn, may add further information to a screening, indicating whether specific issues have been resolved via consultation.

## 4. Methods

This section covers key elements of the platform implementation. The platform itself consists of various docker containers that are properly orchestrated through the docker engine itself. As stated before, for example, a ready-to-use KeyCloak container for authentication management is used. Other components, like the dedicated databases or the server, are also bundled using a docker.

### 4.1. FeelBack Software Architecture

Figure 3 shows the developed architecture at a very high level. The server itself comprises of a backend implementing the respective business logic. For example, various instruments can be uploaded to the server, or different organizations can be created and maintained. Further, screening data are attached to a particular instrument and patient. This backend, in turn, has its own database, where collected screening data are stored. Most importantly, this database does not store any personal information of the users (i.e., login credentials or e-mail addresses). To secure the backend service, a dedicated authentication service is set up and configured. For this purpose, we added a *KeyCloak* docker image, a state-of-the-art authentication solution, which also handles authorization and role-based access [58]. All user-dependent information (i.e., login credentials) are stored within the KeyCloak database to provide a more secure approach for data management itself. In order to allow for an effective communication with client applications, a sophisticated *GraphQL* API was designed and implemented. This API, in turn, provides *Queries* and *Modifications* to read and write data. Access to respective API is secured via *JWT* (JSON Web Tokens; https://jwt.io/; last accessed: 23 October 2020), a state-of-the-art token-based authentication mechanism [59]. The developed API may be enhanced later to also provide an HL7 export. Note that HL7 FHIR also explicitly mentions GraphQL endpoints, so this concept may not be an issue. However, the developed API can easily be extended by a RESTful API, thereby complying to the FAIR principles [60].

Moreover, we implemented various client applications to communicate and interact with the described server component. For example, we developed a mobile application used to collect data from cancer patients or family members. This application, in turn, is mainly used to load screening instruments from the server, process the latter, and the send collected data back to the server application. Furthermore, we also developed a monitoring application that is used by involved health care professionals. Staff members are able to access a personalized dashboard, showing recent filled in screening instruments from their patients (i.e., depending on their role within the platform). Additionally, historic information of patients are visualized in order to indicate trends in psychological treatment. Finally, screening rules (i.e., cut-off values) are automatically evaluated and shown to medical personnel to highlight critical issues with patients. All of our concepts are designed under the aspects of the major topics of different guidelines for mHealth applications, such as usability, privacy, security, among others [49,50,51,52].

### 4.2. Technical Implementation of the FeelBack Application

All major parts of the platform are developed with TypeScript, which is a strongly typed version of JavaScript. Respective language, in turn, offers well-established features that are known from other object-oriented programming languages, like classes, interfaces, or generics. All of the developed clients (i.e., the mobile data collection application and the monitoring application) rely on Angular and Ionic, respectively. Ionic is an established cross platform development framework, allowing to write application code using web technologies (i.e., HTML, CSS and JavaScript). However, this web application is then wrapped in a specific container that can be executed on various mobile operating systems, like Android or iOS. This container also offers specific APIs in order to communicate with the device itself, like accessing sensors (camera, GPS, …) or using the file system [61]. Figure 4 illustrates the user interface of the developed mobile application. Note that this user interface is responsive and it automatically adapts to specific device types to a certain degree. For example, the user interface may be restructured when executed on a tablet computer instead of a smartphone.

It is noteworthy that the mobile application itself does not require a login via e-mail and password. In interviews with potential users and medical staff, we found out that most of the patients would not want to use such a credentials-based login. This may be because they do not want to sign up via their e-mail address; some of them mentioned concerns regarding not being anonymous anymore. Other users did not have an e-mail address at all. With this additional information in mind, we designed an anonymous ID card containing a QR code. This code, in turn, contains respective credentials of a patient. Each participating organization (i.e., a clinic or counseling center) orders some of these ID cards and hands them out to patients during their first consultation. The mobile application, in turn, is able to scan this code and attaches filled in screening instruments to this particular pseudonym.

The monitoring application allows medical staff to review uploaded screening data from their patients (cf. Figure 5). Psychologists can filter results that are based on particular instruments (i.e., only Distress Thermometer data for a specified patient). Subsequently, a dashboard is presented that shows the treatment progress of this particular patient (i.e., a line diagram indicating their distress over time). Psychologists can drill down on this data to review one screening filled in by this person. Thereby, not only the original input data can be reviewed, but charts are also automatically generated by the application. The rules that are used to generate the diagrams depend on respective instrument. For the Distress Thermometer, for example, the line chart is calculated from the value distress from the instrument. The radar chart is calculated applying aggregation functions on the problems list. Of course, other instruments may provide more sophisticated evaluation rules that may also be automatically incorporated in the monitoring application.

A controlled feasibility study was conducted in order to obtain information on the feasibility and applicability of the platform developed. An outpatient setting in a local cancer counseling center was chosen for this assessment.

### 4.3. Evaluation of the Pilot Feasibility Study

Figure 6 shows the procedure of the study. When recruiting participants for this study, we tried to couple the study with a visit at the cancer counseling center in Ulm. In all cases, participants (i.e., patients or relatives) were interviewed right before their regular counseling visit. The latter were also informed that the participation is not required to obtain their psychological consultation. First, participants were greeted and obtained detailed information regarding the study. For example, they became informed about the goal of the study or the time they have to invest. Next, they got handed out the consent form for this study, which they have to sign in order to participate. If a participant decides to not sign the form (or did not want to participate for whatever other reason), then he/she was asked to fill in a non-participation questionnaire to indicate the reasons (i.e., not interested, too much stress). Of course, the participants still get consultation even if they did not participate in the study. However, if they sign the consent form, they had to fill in a demographic questionnaire providing some basic personal details (i.e., gender, age, …). Further, they are asked to fill in the Distress Thermometer, which is a de-facto gold standard validated approach to assess psychological distress in oncological care [25,62]. Based on randomization, some of the participants had to process the psychological instrument on paper, whereas the others had to process a digital version on a provided tablet computer. Finally, all of the participants were given a form assessing their own view on the user-friendliness of the Distress Thermometer itself, the acceptance of a digital vs. paper-based screening version, as well as questions regarding the attitude towards potential cross-sectional data exchange in the future. In the context of this study, only the results of the acceptance of a digital screenings are reported. The overall procedure took about 20 min, in total. Afterwards, the regular psychological consultation started.

To assess the acceptance of the digital screening procedure, we adopted four acceptance indicators from a previous study that investigated the acceptance of a standardized digital screening in patients with breast cancer [63]. While this is not a validated questionnaire, it was successfully applied in other studies [63]. For the rating a bipolar likert-scale was used, ranging from 1 to 10. The variables are *Suitability* (less suitable (1) to more suitable (10)), *Strenuousness* (more strenuous (1) to less strenuous (10)), *Difficulty* (more difficult (1) to less difficult (10)), and *Preference* (no preference (1) to absolute preference (10)). Respective questions as well as possible answers were translated to English for the sake of understandability in the context of this manuscript are shown in Figure 7. Participants working with the paper-based version and did not have the hands-on experience with the digital version were still asked to give their opinion on how they generally would like it to use a digital instrument.

Two of the acceptance variables mentioned have a reversed, negative connotation (i.e., *Strenuousness*, *Difficulty*), whereas higher values on the rating scale indicate favoring the digital instrument. To provide a better understanding of the results, these two items will be referenced as positive terms (i.e., *Convenience*, *Ease of use*) in the following. The variable *Overall Acceptance* that was reported in this manuscript was not part of the questionnaire, but was calculated later by summarizing the individual acceptance variables. Respective psychometric statistics are reported in Section 4.5. The Ethics Committee at Ulm University approved all of the materials and methods used in the context of this pilot feasibility study. The latter was carried out in accordance with these approved guidelines. All of the study participants gave their informed consent.

### 4.4. Participants for the FeelBack Pilot Feasibility Study

In total, 61 individuals were recruited at the cancer counseling center in Ulm and interviewed in this first pilot feasibility study. Thereby, one participant was excluded due to not meeting inclusion criteria. People willing to participate in our study were carefully instructed according to the study design. All of the material was handed out in German to ensure that all participants understood respective questions [64]. 58% of participants included were cancer patients. The age ranged from 23 to 77 with a mean of 53.1 (standard deviation (SD) = 12.87) years. 75% were female, representing the commonly observed gender ratio of people seeking support in the field of psycho-oncology [65,66]. The psychological distress of participants was rather high with an average of 6.42 (SD=2.13). At the start of the study, the participants were randomly assigned to one of the two groups; working with the digital or the traditional paper-based screening. Table 1 displays descriptive statistics for relevant variables.

### 4.5. Statistics

Data preprocessing showed that there were neither substantially nor systematically missing data. Group differences in descriptive statistics were analyzed using Fisher’s exact test for nominal variables and parametric two-tailed *t-test* for metric/ordinal variables. For the calculation of the overall acceptance variable psychometric statistics for measuring reliability were calculated. Intercorrelations of the four indicator variables using Pearson correlation were moderate to high (ranging between r = 0.532 and r = 0.795), and they were significant on α<0.01 level. Cronbach’s α shows good internal consistency (α=0.864). The group differences in acceptance variables were tested using parametric two-tailed *t-test*. Prior to parametric tests, data were checked for normal distribution and equality of variance. Whenever normal distribution was not given, calculations have additionally been performed using a non-parametric Mann–Whitney U test. As results remained stable with both tests and simulation studies showed that the independent samples *t-test* is robust against violations of the normal distribution assumption [67] (especially if n1,2≥30), only *t-test* results are reported. The correlation of acceptance variables with age were calculated using Pearson correlation coefficient.

### 4.6. Data Availability

The underlying raw data set containing all data collected in the context of this pilot feasibility study is included in this manuscript (and its Appendix A).

## 5. Results

The main aim of our study is to assess the feasibility of introducing a digital application in monitoring the mental health status of oncological patients. This feasibility is assessed in a control pilot evaluation of 60 participants that were allocated to two groups: the paper-based questionnaires and the digital approach. First, we assessed the acceptance screening instruments (*Suitability*, *Convenience*, *Ease of use*, and *Preference*) in general. Second, we investigated the differences between the two groups concerning the acceptance of the used instruments. Altogether, we could show an increase of the overall acceptance by 58.5% in the mean when using a digital screening as compared to the paper-based. In addition, we found significant differences among the two groups in all four acceptance criteria. In the following, we show the results evaluating the acceptance of our digital platform FeelBack in more detail.

### 5.1. Baseline Comparison

Table 1 summarizes the sample description and comparison between the two different screening types (i.e., paper vs digital). The results show that there were no significant group differences in the distribution of advice seekers ($\chi^{2}=0.002$, *p* = 1.0), gender ($\chi^{2}$ = 0.2, *p* = 0.769), migration background ($\chi^{2}=0.246$, *p* = 0.702), age (t (58) = 1.25, *p* = 0.217), highest education ($\chi^{2}=4.61$, *p* = 0.222), current employment ($\chi^{2}=0.003$, *p* = 1.0), and distress value (t (59) = 0.73, *p* = 0.466), indicating that descriptive variables are likely not to explain differences in acceptance ratings.

### 5.2. Results for Research Questions

#### 5.2.1. Do Participants Accept Screening Instruments in Psycho-Oncology?

The first research question examined the acceptance of screening instrument in psycho-oncology in general. Table 2 shows the mean values of assessed acceptance variables (i.e., *Suitability*, *Convenience*, *Ease of use*, *Preference*), ranging from 6.20 to 6.68, as well as the *Overall Acceptance* variable with a mean of 6.42. Thereby, values toward 0 indicate a low acceptance rating, whereas values towards 10 represent high acceptance ratings. These results indicate a trend towards the digital version when compared to the paper-based one in our study sample. In particular, personal *Preference* of a digital screening instrument was rated high by participants (M=6.68,SD=2.83). The biggest variance was observed in *Suitability* (M=6.57,SD=3.01), indicating a slightly wider distribution of values on our rating scale.

#### 5.2.2. Are There Any Differences with Respect to the Acceptance of Instruments between the Groups Using the Digital versus the Paper-Based One?

The second research question asked whether there would differences in the groups using either the digital or the paper-based instrument with regard to acceptance, convenience, ease of use, and preference. Between-group analyses revealed significant group differences. We quantified the inter-group differences between digital screening and paper-based screening results, as shown in Table 3 and Figure 8. The results show highly significant group differences as well as high effect sizes (Cohen’s d) in all four indicator variables of acceptance (*Suitability* (*t* (58) = −3.64, *p* = 0.001, d = 0.94), *Convenience* (t (57) = −4.52, *p* < 0.001, d = 1.18), *Ease of use* (t (58) = −5.10, *p* < 0.001, d = 1.32), *Preference* (t (58) = −5.33, *p* = < 0.001, d = 1.38), as well as for *Overall Acceptance* (t (58) = −5.89, *p* < 0.001, d = 1.52). The participants who used the digital screening rated a future digital version significantly more suitable, more convenient, easier to use, and more preferable than participants who lacked the practical experience in the study trial. However, participants using the paper-based screening still showed moderate acceptance that can be interpreted as being neutral and open for digital innovation. In addition, we checked on possible influences on age on *Acceptance* ratings. In this context we count small negative non-significant correlations between age and *Overall Acceptance* (r=−0.20,p=0.132), *Suitability* (r=−0.24,p=0.071), *Convenience* (r=−0.09,p=0.513), *Ease of use* (r=−0.13,p=0.318), and *Preference* (r=−0.22,p=0.099).

## 6. Discussion

The results of this pilot feasibility study demonstrated that a digital screening for psychological distress of cancer-patients and their relatives is well-accepted and even preferred over a paper-based version. These findings correspond with previously conducted studies in the field of psycho-oncology [63,68,69,70,71]. Furthermore, the results of this study show that the mobile application, including the Distress Thermometer as the screening instrument, was met with approval and considered to be *suitable*, *convenient*, and *easy to use*. These results are a fundamental pillar for future implementations of the mobile application in the course of digitization strategies within the medical domain and psycho-oncology respectively.

As an interesting result, the participants who got to experience the use of the digital instrument showed higher *Acceptance* than participants not having this experience. This is in line with results of other studies showing increased acceptance after introduction of digital material [72]. It can be assumed that prior doubts or reservations might be a reason for this. Because cancer patients and relatives experience significant emotional stress in the course of the disease, as well as spend a lot of time with treatments and bureaucratic demands [7], it is desirable to reduce additional strain during treatment as much as possible. However, our findings show that a digital screening is not considered more, but rather less strenuous. Therefore, dismissive attitudes might have become less prevalent with actual use, which has also been shown in a previous study [63]. In conclusion, the implementation of a digital screening can make the procedure of regular screenings more economically efficient without increasing the stress that is experienced by cancer patients or relatives.

Supporting these findings, there were no significant group differences in the reported distress values or in age between the groups, potentially explaining the differences in the *Overall Acceptance*. It has been found that an increasing age can have a negative influence on the willingness to use and the experience with digital instruments [73,74]. Indeed, we did find negative relations between age and all acceptance variables. However, the correlations were consistently small and not significant, allowing for the conclusion that age did not play a significant role. Because, in our sample, the average age was 53.1, ranging up to 77; this further supports the feasibility of a digital screening, even in populations with a higher age.

## 7. Conclusions

mHealth applications offer the potential to reduce communication overhead and optimize treatment strategies. The FeelBack platform aims to provide a valid alternative to the standard paper-based screening of the stress condition of cancer patients and their relatives. In the context of psycho-oncology, paper-based questionnaires, such as the Distress Thermometer, are frequently used. However, paper-based evaluation is error-prone and time consuming to digitize and, thus, using computer-tools for data analysis is limited. Nevertheless, it is not yet clear whether these digital approaches are accepted by cancer patients and their relatives. Our pilot feasibility study of the FeelBack platform aimed to investigate the acceptance of such a platform when compared to the standard paper-based screening. The FeelBack platform outperformed the traditional approach on all measured acceptance criteria, such as *Suitability*, *Ease of use*, and *Convenience*, among others. This result is even more encouraging when considering the broad distribution of participants according to their age. Finally, the results of the pilot feasibility study support the further usage of such a platform in the field of psycho-oncology.

## Figures and Tables

**Figure 1 ijerph-18-05092-f001:**
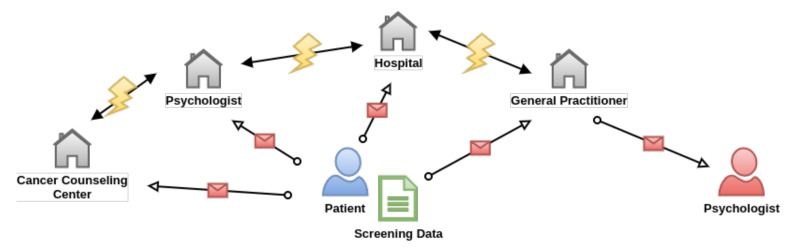
As-Is State of the Application Scenario. The figure depicts the current scenario of data transfer in psycho-oncological treatment. The patient has to submit a paper-based questionnaire to each involved health care provider. These different providers usually do not share data across each other due to the lack of direct communication. Consequently, data are not synchronized and are prone to be incomplete for certain actors.

**Figure 2 ijerph-18-05092-f002:**
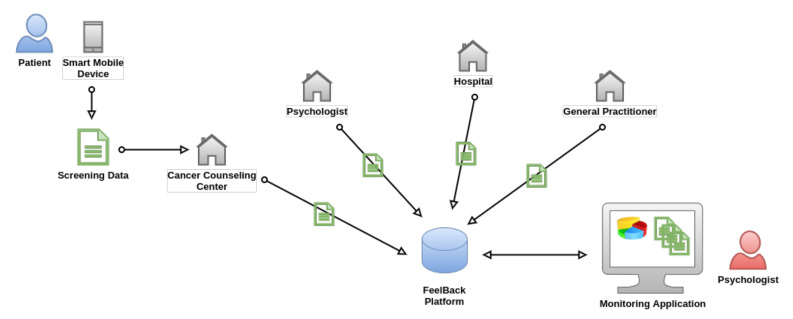
To-Be State of the Application Scenario. The FeelBack-platform allows for centralizing patient information in the context of psychological distress. After registration, the data collected are independent of the involved health care provider. The data are directly transmitted to a centralized hub (FeelBack platform) in a pseudonymous way. In addition, involved health care providers can add data to health record of the corresponding patient at the centralized hub. This platform allows psychologists to access data that are always synchronous and up-to-date via a monitoring application.

**Figure 3 ijerph-18-05092-f003:**
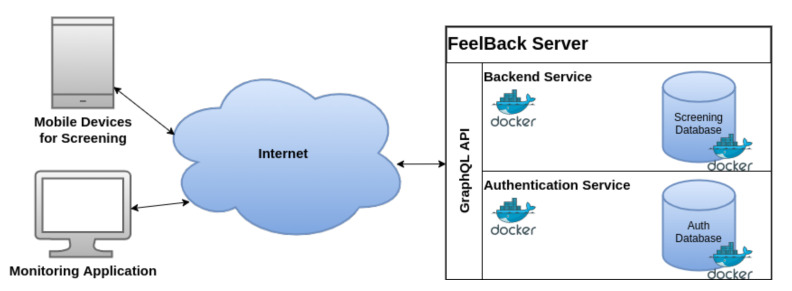
Architecture of the FeelBack-platform. Both, screening devices and monitoring application access the FeelBack server via Internet. The FeelBack-server comprises a dockerized authentication service to secure the server via *KeyCloak*. The screening data per-se are stored separately and do not contain any personal information of the users.

**Figure 4 ijerph-18-05092-f004:**
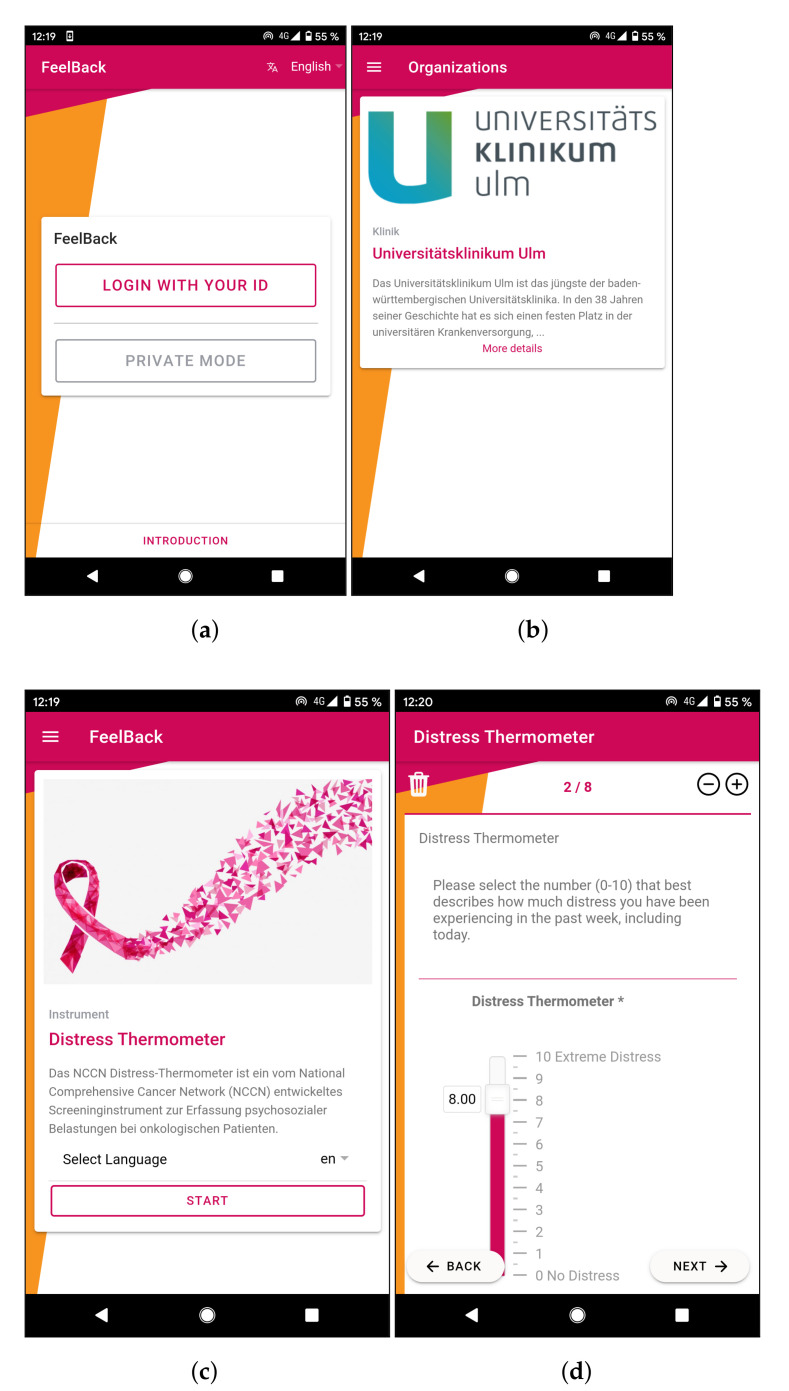
Screenshots of the Mobile Data Collection Application within the FeelBack platform. (**a**) Shows the login screen of the FeelBack application. Users can access the application via their respective ID. (**b**) Depicts an exemplary overview about the registered organizations. Here, it is showing general information and contact details for Ulm University Hospital as health care provider. (**c**) A screenshot of the home screen of the FeelBack application. From here, a user can access settings via the upper menu or start to report the current psychological condition via a implemented questionnaire. (**d**) Here, the Distress Thermometer by the National Comprehensive Cancer Network is used. The amount of distress can be entered via answering the respective questions.

**Figure 5 ijerph-18-05092-f005:**
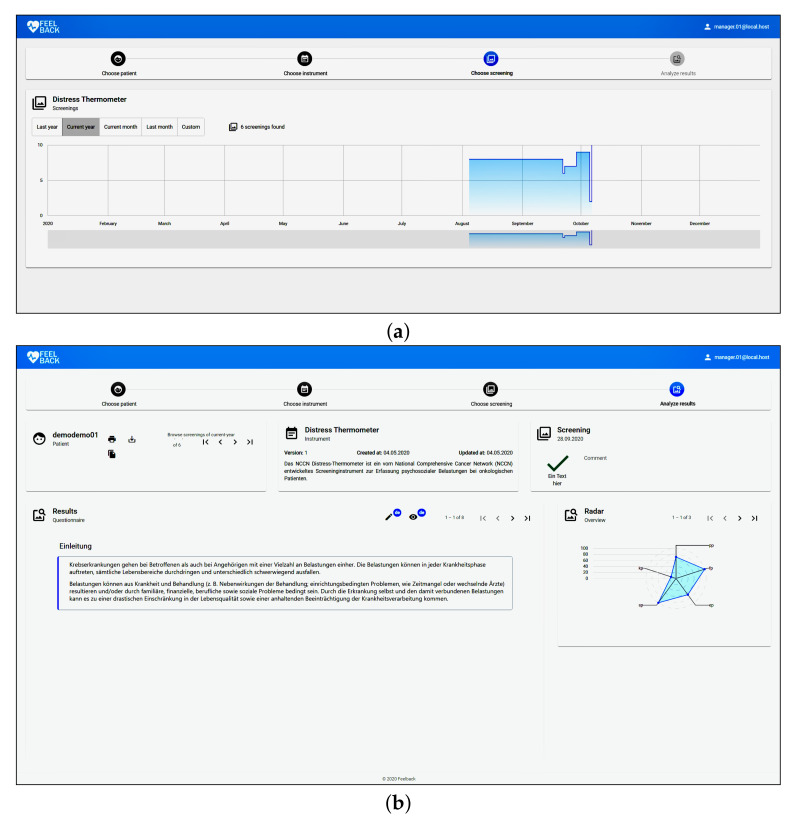
Screenshots of the monitoring application. (**a**) Shows the psychological distress for one particular patient across time to provide a high-level overview for the health care provider. (**b**) Shows the data for one particular screening entry. More precisely the psychologist can review given answers and get visual feedback about the patients psychological condition at that time (i.e., emotional problems, personal problems, etc.).

**Figure 6 ijerph-18-05092-f006:**
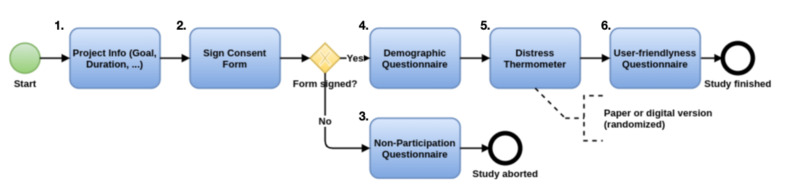
Process diagram of the pilot study procedure. The goal of the study was introduced (step 1, blue boxes) and individuals had to sign a consent form (2). If consent was not provided, individuals were asked to fill in a form regarding why they would not like to participate (3) and then had their regular counseling hour. If consent was provided, they were asked to provide demographic information (4) and fill in the Distress Thermometer (5) either on paper or on the tablet (choice was randomized). Finally, they had to give feedback regarding our four evaluation criteria (6), before they moved on with the regular counseling hour.

**Figure 7 ijerph-18-05092-f007:**
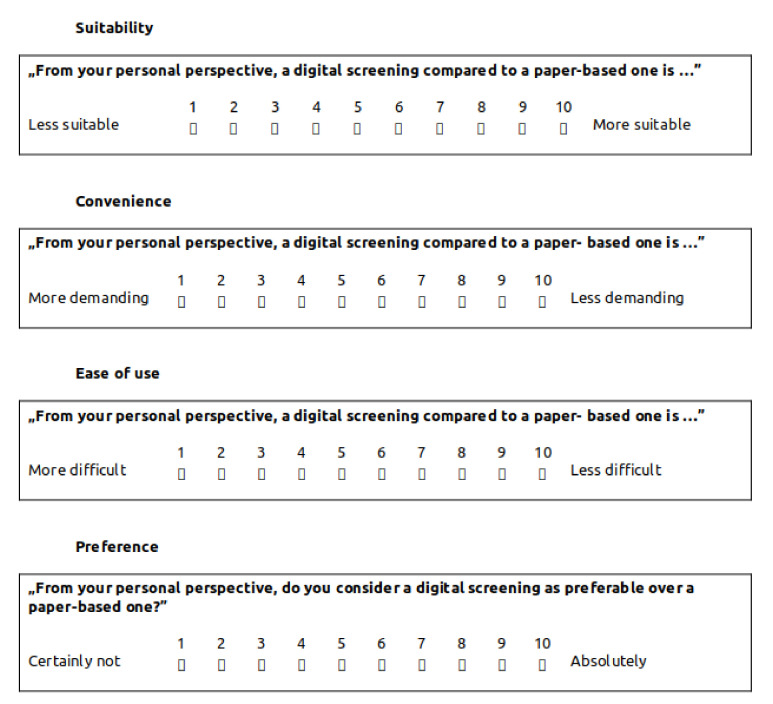
Questionnaire addressing the feasibility of the digital screening versus the paper-based one. Suitability, convenience, ease of use and preference of digital screening compared to paper-based screening are evaluated via this questionnaire. All questions can be answered on a bipolar likert-scale ranging from 1 to 10. Original questions are formulated in German, being translated to English for the sake of comprehension.

**Figure 8 ijerph-18-05092-f008:**
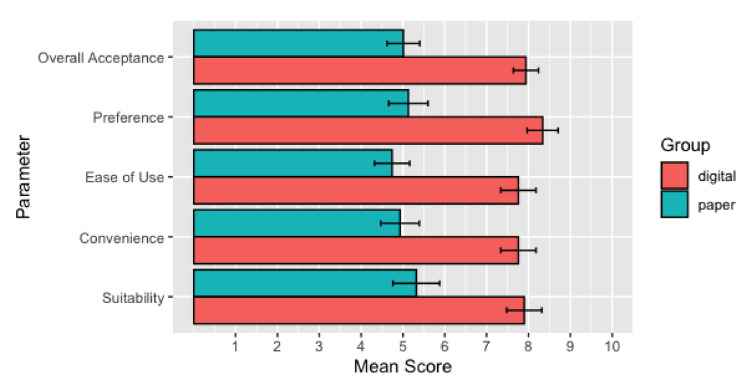
Group comparison of acceptance criteria of a digital compared to a traditional paper-based screening for accessing psycho-social distress in cancer patients and relatives on a scale from 0 (bad) to 10 (good). The study participants were randomly allocated to groups either processing a digital screening on a smart mobile device or a traditional paper-based screening. All group differences are statistically significant (p<0.01) and error bars in the plot describe standard errors.

**Table 1 ijerph-18-05092-t001:** Sample descriptions and comparisons between screening trials in baseline variables.

Variable	Paper-Based Screening (N=31)	Digital Screening (N=29)	*p*-Value
Advice Seeker, n (%)			p=1.000 ^a^
Patient	18 (58)	17 (59)	
Relative	13 (42)	12 (41)	
Gender, n (%)			p=0.769 ^a^
Female	24 (77)	21 (72)	
Male	7 (23)	8 (28)	
Migration Background, n (%)			p=0.702 ^a^
No	28 (90)	25 (86)	
Yes	3 (10)	4 (14)	
Age (years), mean (SD)	55.1 (12.2)	51.0 (13.4)	p=0.217 ^b^
Highest education, n (%)			p=0.222 ^a^
Apprenticeship	12 (39)	13 (45)	
Professional School	1 (3)	4 (14)	
University	13 (42)	11 (38)	
Other	5 (16)	1 (3)	
Currently Employed, n (%) ^1^			p=1.000 ^a^
No	12 (39)	11 (38)	
Yes	18 (58)	17 (59)	
Distress value, mean (SD)	6.6 (2.1)	6.2 (2.2)	p=0.466 ^b^

^1^ = *N* = 58/60 participants (96%; 2 missing values); ^a^ = Fisher’s exact test; ^b^ = Independent samples *t*-test.

**Table 2 ijerph-18-05092-t002:** The acceptance of screening instruments on a bipolar likert-scale ranging from 1 (bad) to 10 (good).

Parameter	Mean	SD
Suitability	6.57	3.01
Convenience	6.32	2.78
Ease of Use	6.20	2.74
Preference	6.68	2.83
Overall Acceptance	6.42	2.42

**Table 3 ijerph-18-05092-t003:** Group comparision of acceptance between using a digital screening (FeelBack application) and traditional paper-based screening.

Variable	Groups		Statistics	*p*-Value ^a^	Cohen’s *d*
Paper-Based Screening (N=31) Digital Screening (N=29)					
Suitability	5.32 (3.11)	7.90 (2.27)	t(58)=−3.64	p<0.01	0.94
Convenience ^b^	4.93 (2.52)	7.76 (2.28)	t(57)=−4.52	p<0.001	1.18
Ease of Use	4.74 (2.34)	7.76 (2.25)	t(58)=−5.10	p<0.001	1.32
Preference	5.13 (2.60)	8.34 (2.00)	t(58)=−5.33	p<0.001	1.38
Overall Acceptance	5.01 (2.19)	7.94 (1.61)	t(58)=−5.89	p<0.001	1.52

Values are given as mean values (± SD) referring to a bipolar likert-scale from 0 (e.g., less convenient) to 10 (e.g., more convenient). *Overall Acceptance* was calculated by summarizing the given variables. ^a^ = *p*-values of group differences are calculated using independent two-tailed *t*-test; ^b^ = *n* = 59; one participant of the paper-based screening group did not respond to this item.

## Data Availability

All data is available in the manuscript or its Appendix A.

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
