# Peer review of "Supporting Medical Staff from Psycho-Oncology with Smart Mobile Devices: Insights into the Development Process and First Results"

_ijerph, 2021, doi:10.3390/ijerph18105092_

Round 1
Reviewer 1 Report
Authors tried to show that exchange of such psychological information will ultimatively help the patients to get a better (i.e., more suitable) treatment to deal with identified issuesvia digital devices. The way the resarch presented must be improved. Introduction is too long and there are lots of needless informations to present this data compared to insufficient discussion. Contents and style must be changed dramatically. It is harad to understand what is the most important findings authors want to present in this paper.
Reviewer 2 Report
The abstract could include some details of methods such as the number of subjects and the setting.
Reviewer 3 Report
In the era of universal digitalization (and pandemic Covid 19), we more and more often use technological achievements to provide assistance and monitor the health of patients, also in psychological care giving. Therefore, research in this area is as up-to-date and necessary as possible.The reviewed manuscript analyzed the usefulness of mobile application in helping onological patients - “The FeelBack platform”. I think that such digital projects are the future of psychological care so I recommend this paper for publication, however, some disadvantages of the manuscript need to be consider before.
- The abstract only briefly describes the study and suggests more sophisticated experiment with monitoring (longitudinal data) and comparing/aggragating the data from digital mobile application and medical experts.
- Some of the titles and subtitles of the sections (2;3; 3.2; 3.3.; 4; 5.3.) have a form of shortcuts and should more precisely described to be consistent with the content and understandable for readers
- The section “2.1. Related Work in the Context of Mobile Data Collection and mHealth Applications” is rather a general description of the problem without detailed indications of the advantages, but especially the dangers of using such applications in medicine
- In section 2.2. it would be good to add existing in the literature guidelines for mobile applications
- The authors used only one psychological instrument: Distress Thermometer that is not a questionnaire, the second method named by the authors User Friendliness Questionnaire consist only 4 items and was not properly tested in order of its psychometric properties (additionally there was not theoretical background why specifically this questions were asked and not different). Please add proper psychometric information and calculate proper statistics to prove that it is q valuable questionnaire
- In section results all tables should be described – in section 6.1. the baseline comparisons -the results of t statistics are not described
- The data Overall Acceptance was based on a simple summarize of four variables (i.e., Suitability, Convenience, Ease of use, Preference) please add reliability of this construct and it would be good to show intercorrelations between 4 variables as you only theoretically assumed that they are components of one overall construct
- The group of participants that gave hypotheticall opinion while using the established paper-based version in not clearly defined – add information what they have judged – the paper – pencil version once more or the digital version.
- One of the limitsation of the study is not use the longitudinal paradigm: comparing the difference within group of digital application i.e. the overall acceptance at the beginning and the end of the project? Especially, as although digital application may be more preferable because of its novelty effect at first time the attitude may change with time of its usage.
- The summary although contain interesting information are not conclusions of the findings.
